# Profiling Physical Fitness of Physical Education Majors Using Unsupervised Machine Learning

**DOI:** 10.3390/ijerph20010146

**Published:** 2022-12-22

**Authors:** Diego A. Bonilla, Isabel A. Sánchez-Rojas, Darío Mendoza-Romero, Yurany Moreno, Jana Kočí, Luis M. Gómez-Miranda, Daniel Rojas-Valverde, Jorge L. Petro, Richard B. Kreider

**Affiliations:** 1Research Division, Dynamical Business & Science Society—DBSS International SAS, Bogotá 110311, Colombia; 2Research Group in Biochemistry and Molecular Biology, Universidad Distrital Francisco José de Caldas, Bogotá 110311, Colombia; 3Research Group in Physical Activity, Sports and Health Sciences (GICAFS), Universidad de Córdoba, Montería 230002, Colombia; 4Sport Genomics Research Group, Department of Genetics, Physical Anthropology and Animal Physiology, Faculty of Science and Technology, University of the Basque Country (UPV/EHU), 48940 Leioa, Spain; 5Grupo de Investigación Ciencias Aplicadas al Ejercicio, Deporte y Salud—GICAEDS, Universidad Santo Tomás, Bogotá 205070, Colombia; 6Department of Education, Faculty of Education, Charles University, 11636 Prague, Czech Republic; 7Sports Faculty, Autonomous University of Baja California, Tijuana 22390, Mexico; 8Núcleo de Estudios para el Alto Rendimiento y la Salud (NARS-CIDISAD), Escuela Ciencia del Movimiento Humano y Calidad de Vida (CIEMHCAVI), Universidad Nacional, Heredia 863000, Costa Rica; 9Clínica de Lesiones Deportivas (Rehab&Readapt), Escuela Ciencia del Movimiento Humano y Calidad de Vida (CIEMHCAVI), Universidad Nacional, Heredia 863000, Costa Rica; 10Exercise & Sport Nutrition Laboratory, Human Clinical Research Facility, Texas A&M University, College Station, TX 77843, USA

**Keywords:** cardiorespiratory fitness, physical endurance, muscle power, sprint speed, range of motion, unsupervised machine learning

## Abstract

The academic curriculum has shown to promote sedentary behavior in college students. This study aimed to profile the physical fitness of physical education majors using unsupervised machine learning and to identify the differences between sexes, academic years, socioeconomic strata, and the generated profiles. A total of 542 healthy and physically active students (445 males, 97 females; 19.8 [2.2] years; 66.0 [10.3] kg; 169.5 [7.8] cm) participated in this cross-sectional study. Their indirect VO_2max_ (Cooper and Shuttle-Run 20 m tests), lower-limb power (horizontal jump), sprint (30 m), agility (shuttle run), and flexibility (sit-and-reach) were assessed. The participants were profiled using clustering algorithms after setting the optimal number of clusters through an internal validation using R packages. Non-parametric tests were used to identify the differences (*p* < 0.05). The higher percentage of the population were freshmen (51.4%) and middle-income (64.0%) students. Seniors and juniors showed a better physical fitness than first-year students. No significant differences were found between their socioeconomic strata (*p* > 0.05). Two profiles were identified using hierarchical clustering (Cluster 1 = 318 vs. Cluster 2 = 224). The matching analysis revealed that physical fitness explained the variation in the data, with Cluster 2 as a sex-independent and more physically fit group. All variables differed significantly between the sexes (except the body mass index [*p* = 0.218]) and the generated profiles (except stature [*p* = 0.559] and flexibility [*p* = 0.115]). A multidimensional analysis showed that the body mass, cardiorespiratory fitness, and agility contributed the most to the data variation so that they can be used as profiling variables. This profiling method accurately identified the relevant variables to reinforce exercise recommendations in a low physical performance and overweight majors.

## 1. Introduction

The university or college lifestyle may generate changes in students’ health due to sedentary habits [1]. Some situations that lead to health problems among university students include inadequate nutrition, leisure time, low levels of physical activity, the consumption of psychoactive substances, stress mismanagement, academic and psychosocial pressure, and an insufficient follow-up of medical recommendations [2,3,4]. A high prevalence of sedentary behavior in university students of different majors has been reported which affects the diverse aspects of wellbeing and might result even in depression [5] or being overweight [6]. This in turn modifies the biological and psychological parameters that disrupt the quality of life. Besides the aforementioned factors, the initiation of university life leads to an increase in academic activities, social interaction with peers, and the constant use of electronic devices or screen time; therefore, a significantly reduced physical activity, increased intake of “fast food”, and raised stress levels might predispose this population to the development of chronic diseases [7].

In Colombia, evidence related to sedentary behavior among university students has shown that students spend several hours of their time on electronic devices per day [8]. A systematic review reported that sedentary behavior, especially the sitting time and screen exposure, is positively associated with obesity and other chronic diseases (e.g., dyslipidemia, hypertension, diabetes, etc.) in young Latin American adults [9]. Therefore, wellness programs should be considered to be an elementary component of tertiary education institutions to deepen the aspects of university students’ lifestyles, which directly impacts the lack of activity and increased sedentary behavior [10,11,12,13]. Additionally, it is important to highlight the academic load that students undergo each semester which influence the individual conditions and the leisure spaces. Actually, major students may consider physical activity as irrelevant or unrelated to their health and academic performance, which is contrary to the available evidence [14].

In undergraduate programs related to physical activity or sport (e.g., Physical Education or Sports Sciences majors), there are admission protocols that involve standardized physical tests that, in part, allow for evaluating the level of physical fitness (PF) of the applicants. In this sense, it has been found that PF is adequate in the first year. Still, it substantially deteriorates compared to senior students, which could be attributed to academic dynamics and workload, among other personal factors to which most future graduates are exposed to [2]. Despite the relevance of the PF evaluation process for the detection and control of the risk factors in the university population, monitoring this aspect is not a process that is carried out systematically in university training programs, including those of physical education, sports, or similar majors. Thus, this study aimed to identify the physical fitness profiles of majors enrolled on the Physical Culture program at the Universidad Santo Tomás in Bogotá, Colombia. We believe this reproducible analytical approach would provide valuable information for implementing assertive strategies for health promotion and disease prevention among university students.

## 2. Materials and Methods

### 2.1. Study Design

This cross-sectional study was reported following the Strengthening the Reporting of Observational Studies in Epidemiology—STROBE guidelines [15].

### 2.2. Setting

The study was carried out from February to November 2019. The standardization considered the following recommendations given to the participants to avoid biases in the data collection. Concerning their diet, the participants were instructed that they should not have ingested food two-four hours prior to the evaluation, nor should they have consumed alcohol or coffee eight hours before measuring their body mass. Similarly, they were informed that they should not have consumed corticosteroid or diuretic medications eight days before the assessment. Regarding physical activity/physical exercise, it was insisted that no physical exertion should be performed 24 h before attending the test; in addition, bladder emptying was suggested prior to the data collection. The physical tests were carried out in the San Alberto Magno Campus of the Santo Tomás University (Bogota, Colombia) under the supervision of researchers in physical activity with previous experience in applying these tests. The subjects voluntarily participated and were informed about this research’s protocol and aim. The study was designed following the ethical guidelines of the Declaration of Helsinki [16] and the Resolution 8439 of 1993 of the Ministry of Health of Colombia. The Ethics Committee approved the research protocol of the Universidad Manuela Beltrán (CEI-1705228-27).

### 2.3. Participants

The healthy and physically active undergraduate students of a Physical Education, Sports and Recreation major at a private institution in Bogota, Colombia, were invited to participate. This study belonged to a research project approved and developed in collaboration with other institutions. The selection of the subjects was based on the following inclusion criteria: (i) being enrolled in a Physical Education, Sports and Recreation program; (ii) residing in Bogotá and municipalities of the metropolitan area (above 2600 m above sea level); and (iii) not having any medical history or restriction that would make participation impossible during the execution of the physical tests. All the participants in this study were active undergraduate students of the Physical Culture, Recreation and Sport course at the Universidad Santo Tomás.

### 2.4. Variables

The following continuous variables were measured: the body mass (kg), stature (cm), cardiorespiratory fitness (indirect VO_2max_), muscle power (horizontal jump), sprint speed (30 m), agility (Shuttle Run Agility Test), and flexibility (Sit and Reach). Other variables were also collected from the informed consent (sex and academic year). The socioeconomic stratum was obtained from the Bogotá’s planning department using the official address of the participants’ residence.

### 2.5. Data Sources/Measurement

Bogotá city is 2630 m above sea level with temperatures ranging from 9 to 14 °C. The familiarization and data collection were performed in two sessions. In the first session, recommendations were given for the correct execution of the battery test. In addition, it was explained to the participants that one of the examiners would supervise the movements to avoid compromising technical execution while a second examiner would proceed with a photography collection (for comparison with the standardized images of the test). During the second session, the data collection was made between 07:00 and 10:00 and 18:00 and 21:00 (GMT-5), the times the participants were available. Prior to testing, the participants performed a general warm-up consisting of low-intensity jogging for 10 min. All tests were performed once, except for the horizontal jump, which had two attempts. The following order was maintained during the testing process.

#### 2.5.1. Anthropometry

The body mass data were obtained using the Tanita^®^ SC 331S (Tanita Corporation of America, Inc., Arlington Heights, IL, USA). The participants’ stature was measured to the nearest 0.1 cm with a portable stadiometer SECA^®^ 213 (Medical Scales and Measuring Systems, Hamburg, Germany).

#### 2.5.2. Cardiorespiratory Fitness

The maximum oxygen uptake (VO_2max_), as an indicator of cardiorespiratory fitness, was estimated using field tests: the Cooper Run Test (CRT) and Shuttle Run Test (SRT-20m). The CRT protocol was performed on an athletics track (400 m), marked every 100 m, to obtain the distance each subject covered during the 12 min of the test [17]. For this test, the participants were scheduled in the morning (between 8 and 10 A.M., GMT-5), when the sunlight exposure would not be so intense as to reduce the physical performance and thus biasing the results. For the estimation of VO_2max_ (in mL·kg^−1^·min^−1^), we used the equation 22.351 × distance (km) − 11.288. This was developed by Cooper [17] and validated by Penry et al. [18]. On the other hand, the SRT-20m was carried out on the same schedule, but eight days after the application of the CRT to give the participants time to recover. This test was performed in an open field with a flat and stable surface where two cones were installed with 20 m between them. The participants stood next to one of these cones and covered the distance mentioned above in a round trip. The initial velocity was 8.5 km·h^−1^, and the pace of the race increased by 0.5 km·h^−1^ every minute, indicated by a sound previously explained to the individuals. The test ended once the participant failed to keep up with the speed (i.e., 3 m behind the 20 m line on the audio cue) or was able to complete the stage. With the results of this test, the equation 5.857 × final velocity (km/h) − 19.458 [19] was used for the estimation of the VO_2max_ (in mL·kg^−1^·min^−1^).

#### 2.5.3. Lower-Limb Power

The horizontal jump test without a pre-run was performed on a running track with a 300 cm tape measure placed on the ground; the distance reached from 0 cm to the heel of the foot closest to 0 cm was recorded. The protocol was based on the study of Manouras et al. [20] to evaluate the lower-limb muscle power of the participants. Some attempts were made to become familiar with the test, as established by the committee of experts who created the EUROFIT battery [21]. It is worth noting that the validation of different jumping tests using Alpha Cronbach’s coefficients by Markovic et al. [22] showed high-reliability values for the horizontal jump tests (0.93–0.96). A rest interval of two to three minutes was afforded between the three attempts. The highest recorded value was used for the analysis.

#### 2.5.4. Sprint Speed

The 30 m test in the static position was chosen, which was applied on an athletic track with a flat and consistent surface; cones delimited the distance, and the time data obtained during the execution was recorded (Polar RS100; Polar Electro Oy, Kempele, Finland). The purpose of this test is to evaluate the reaction speed and acceleration of the participant. It is a differentiated test compared to other tests that measure the displacement speed. A level of reliability between 0.88 and 0.95 has been found, which varies according to the age and the terrain used, preferably on an athletic track [23].

#### 2.5.5. Agility

The Shuttle Run Agility Test (SRAT) was applied in an open field, with a delimitation of cones at 9.14 m from the initial line, where the wooden blocks were transported in the respective order of the test. Three changes in direction of 180° were made; the action of transporting and covering this length should be in the shortest possible time (Polar RS100; Polar Electro Oy, Kempele, Finland), according to the physical condition of each subject [24]. This agility test was compared and validated by Kutlu et al. [25] who reported no differences between the SRAT versus other agility tests such as the Illinois protocol, zig-zag test, 30 m, Bosco, and T-drill Agility.

#### 2.5.6. Flexibility

The participants’ flexibility was assessed using the classic Sit and Reach (SR) test. In general, SR tests are valid for estimating the hamstring’s extensibility, although they have a low mean validity for estimating the lumbar’s extensibility [26]. They have a high relative intra-rater reliability (intra-class correlation coefficient of 0.89–0.99) [27].

### 2.6. Study Size

Non-probability convenience sampling was used. After the announcement to participate in this study, only the college students that fulfilled all the inclusion criteria were considered enrolled. A total of 613 students were suitable for eligibility.

### 2.7. Statistical Analysis

Descriptive statistics were expressed as the mean, standard deviation, and 95% confidence intervals (95% CI). The data distribution was plotted using horizontal half-violin diagrams. The data were analyzed with the Mann–Whitney U and Kruskal–Wallis (Bonferroni post hoc) tests to determine the differences between the sexes and between the academic years and socioeconomic strata, respectively. Eta-squared (η^2^) was used to report the magnitude of differences assuming 0.09, 0.14, and >0.22 as a small, medium, and large effect size [28]. As we have performed previously [29,30,31,32], the participants were subdivided into clusters using unsupervised machine learning to identify similar data points (natural groupings) and extract the profile patterns. We used the partitioning around the medoids (PAM) algorithm, also known as k-Medoids clustering which, unlike the k-means algorithm, considers the median as the center of a cluster, thus, it is more robust to noises and outliers [33]. This has enhanced the robustness against the outliers and reduced the noise in the unsupervised machine learning process [33]. Moreover, we performed hierarchical clustering with the bottom-up approach, which provides an easy-to-interpret view of the clustering structure [34]. The number of clusters was determined using 30 criterion algorithms comparing the two-to-ten cluster solutions with the R package ‘*NbClust*’. The internal validation for selecting the clustering method to discuss our results was performed with the ‘*clValid*’ package [35]. To determine the differences between the clusters, we performed a Mann–Whitney U test. The packages ‘*factoextra*’ and ‘*fmsb*’ were used to visualize the clustering results and represent the comparison between the clusters as a spider plot, respectively, within the free software environment for the statistical computing and graphics R v4.0.2 [36]. A significance level of *p <* 0.05 was considered using the IBM SPSS v26 (IBM Corp., Armonk, NY, USA).

## 3. Results

### 3.1. Participants

Seventy-one students did not complete all the battery of the tests and were excluded. Therefore, we report the analysis with data obtained for 542 majors attending from the 1st to 4th year of the physical education major (445 males and 97 females). Differences were found between the participants’ age, body mass, and height by sex, but not in the body mass index (Table 1).

The distribution of the physical fitness variables by sex is represented in Figure 1. According to the results, differences were found in all the study variables by sex (*p* < 0.05). Males showed higher levels of cardiorespiratory fitness, lower limb power, and speed; however, university females showed higher levels of flexibility (Figure 1).

The characteristics of the population by academic year and by socioeconomic strata are shown in Table 2.

As expected, statistically significant age differences were found between the freshmen compared to other students (sophomore, junior, and senior). Significantly lower CRT distance covered and CRT VO_2max_ values were seen in seniors against freshman and sophomore students. In the case of the SRT-20M final stage velocity and SRT-20M VO_2max_, there were differences between freshman and junior, sophomore and junior, and junior and sophomore. The 30 m sprint showed lower values in sophomores and seniors compared to freshmen. Although statistically non-significant, the SRAT revealed that juniors (third year students) were less agile than other students. When data are grouped by socioeconomic strata, a higher age was observed in the students classified in medium versus low strata (Table 2).

### 3.2. Main Results

The *clValid()* function revealed that the hierarchical clustering analysis was more appropriate to cluster our data than other methods. Two clusters were identified: n = 318 (cluster 1) and n = 224 (cluster 2), as can be seen in Figure 2.

Physical fitness accounts for the variation in the data. In fact, the comparison between the clusters revealed significant differences in all the variables except for the stature and flexibility (Table 3). This allowed for recognizing clustering as a relevant methodology to describe the differences between the identified phenotypes (clusters) and help to better ensure an internal validity. Multidimensional analysis showed that the body mass, agility, and CRT VO_2max_ were the variables that explained most of the variance in the data (Figure 3).

## 4. Discussion

### 4.1. Key Results

This study aimed to profile the physical fitness of undergraduate students of a physical education major using unsupervised machine learning. We were able to identify two physical fitness-based profiles of majors: (i) older, heavier, overweight, less VO_2max_, less lower-limb power, faster, less agile, and less flexible students (Cluster 1); and (ii) younger, taller, normal weight, higher VO_2max_, more lower-limb power, slower, more agile, and more flexible students (Cluster 2). The main results also showed that as the major progress, the cardiorespiratory fitness decreases, especially in the final year (senior) students. Castro et al. (2020) [2] also observed that agility, speed, and flexibility decrease as the academic curricula advances, with an accentuation in the last year. A decreased cardiorespiratory fitness is associated with sedentary habits in college students [37,38]. It is worth noting that speed showed better outcomes in sophomores and seniors compared to freshmen (first-year students). We have previously reported the reference values for indirect VO_2max_ tests (CRT and SRT-20m) at a high altitude—between 2600 and 3700 m above sea level—because we are aware of the incidence of this environmental condition on the oxygen supply, oxygen utilization, air density, as well as the cellular response [39]. The participants of this study are within this context since a high altitude is considered to be those locations between 2000 to 5000 m above the sea level [40]. Our results also revealed that female participants had a lower cardiorespiratory fitness when compared to males (Figure 1). This significant difference could be partially explained by differences in the serum testosterone concentration if we consider the documented erythrogenic effects of this hormone [41].

Interestingly, the body mass, agility, and cardiovascular fitness explained most of the data variance among the generated profiles (clusters). This might account for the phenotypic differences. We recently reported that applicants to a physical education major have a normal body composition and body mass index values, which indicates that the nutritional status in the first academic years is appropriate for the specific energy needs of the major (which includes heavy physical training) [32]. Notwithstanding, further research should evaluate if the decreased physical condition that we found in senior students may be associated with alterations in the body composition and health-related parameters. Nowadays, it is clear that an unhealthy lifestyle (e.g., overweight/obese phenotype) can trigger long-term health problems and chronic diseases [42,43,44]. Moreover, Lee and Kim [5] highlighted that a sedentary lifestyle increases anxiety, stress, and depression in university and college students.

Complementary, there is a significant association between a low physical fitness and a poor academic performance [45]; therefore, it becomes important to promote physical and recreational activities during university life. In fact, pre-established university activities, facilities, and the accurate identification of students at a higher risk are important factors that might positively impact the physical fitness level, quality of life, and academic performance of major students [46]. Here, we presented a methodological and analytical workflow to profile exercise and sports major students using unsupervised machine learning. This data analysis approach goes beyond the classical descriptive statistical analysis and is based on various algorithms that can be chosen depending on the nature of the data [47]. Our research group used this methodology to generate the profiles of young athletes and college students based on morphological and functional data [29,30,31]. These analytical methodologies are highly reproducible by the scientific community and help to extract further information from a given dataset that might be used to explain the contextualized phenomena in different ways [29]. Concerning the application of this data analysis approach in universities, we encourage human resources and health departments to become familiarized with and apply more frequently unsupervised machine learning algorithms to identify profiles in college students. An integral framework that can be used for assessing wellbeing in majors is the PERMA+4 [48]. This framework is based on nine building blocks to develop wellbeing: positive emotions, engagement, relationships, meaning, accomplishment, physical health, mindset, environment, and economic security. For more detailed coverage of this model, please refer to Donaldson et al. [49].

Our study has certain limitations that should be considered when drawing practical inferences and setting up future research. First, the small sample size of the female population belonging to the physical education major makes it difficult to analyze this subgroup by ranges of ages or academic semesters. Second, since we did not record the body composition, athletic experience, or academic success variables, more research is needed to find associations with physical fitness in this type of student. Third, observational studies cannot be used to demonstrate causality; therefore, future experimental studies are needed to evaluate the effect of modifying variables that explain most of the variation in our clusters.

### 4.2. Interpretation and Generalizability

This observational study provides information regarding the physical fitness of physical education majors residing in a high altitude (above 2600 m above sea level). It also reports potential variables that account for data variation between the majors’ profiles (body mass, agility, and cardiorespiratory fitness). Finally, this study also contributes to the analytical procedures to profile data under the machine learning paradigm and establish rationale for future research. In fact, applying machine learning to characterize undergraduate populations contributes to the identification of patterns, as well as providing a reproducible, feedback-based, and fast tool to analyze data.

## 5. Conclusions

A decreased physical fitness was found in undergraduate students enrolled in a physical education major as the academic program progressed. This is consistent with the literature, but interestingly, only their speed improved as their college life advanced. In addition, there were no significant differences in the cardiorespiratory fitness, power, agility, or flexibility based on socioeconomic stratification. We identified two significantly different phenotypes representing majors with a poor and good physical performance. The information contained in this article might help universities in the early identification of at-risk students and to develop educational programs to promote healthy lifestyles during college age.

## Figures and Tables

**Figure 1 ijerph-20-00146-f001:**
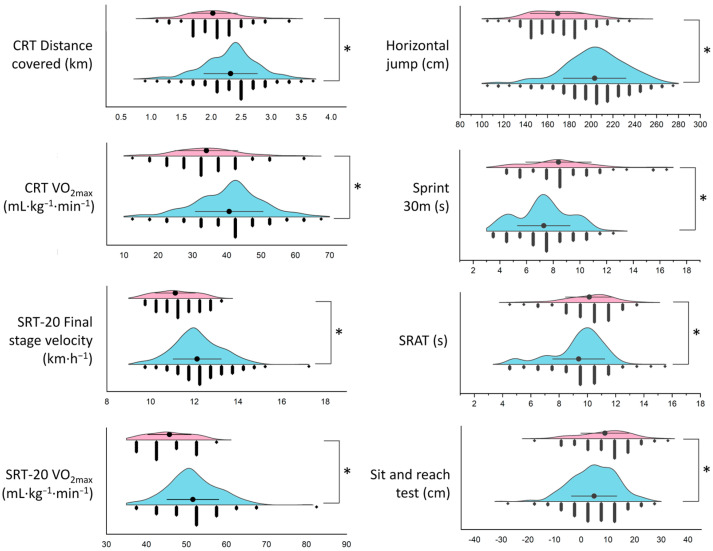
Horizontal half violin plots of the physical fitness variables by sex. This figure shows the distribution of the different physical fitness variables measured in female (pink) and male (blue) majors. Black circles represent the mean, while dark grey diamonds the raw data. CRT: Cooper Run Test; SRAT: Shuttle Run Agility Test; SRT-20M: Shuttle Run Test. * Significant difference *p* < 0.05.

**Figure 2 ijerph-20-00146-f002:**
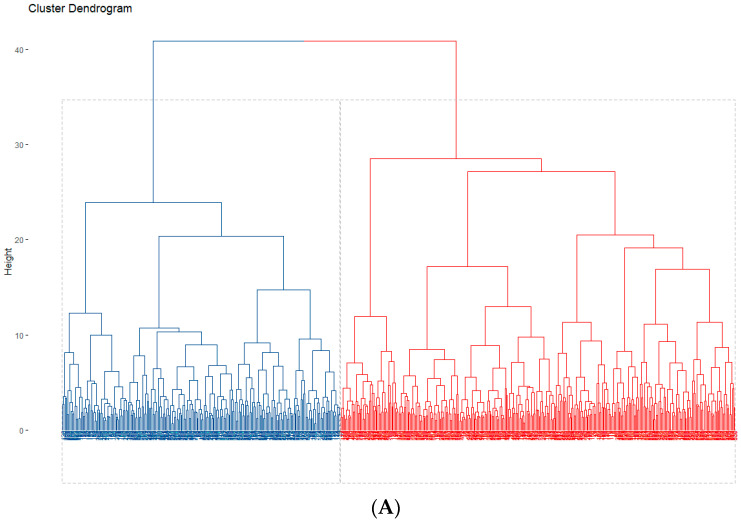
Clustering plots of the analyzed data. (**A**). Dendrogram of the bottom-up agglomerative clustering. Each leaf corresponds to one student. Students who are similar are combined into branches, which are fused at a higher height. The height of the fusion, provided on the vertical axis, indicates the (dis)similarity/distance between two students/clusters. The hierarchical tree was cut to partition the data into clusters (blue for Cluster 1 and red for Cluster 2). (**B**). Cluster plot of the k-Medoids analysis. Red for Cluster 1 and blue for Cluster 2.

**Figure 3 ijerph-20-00146-f003:**
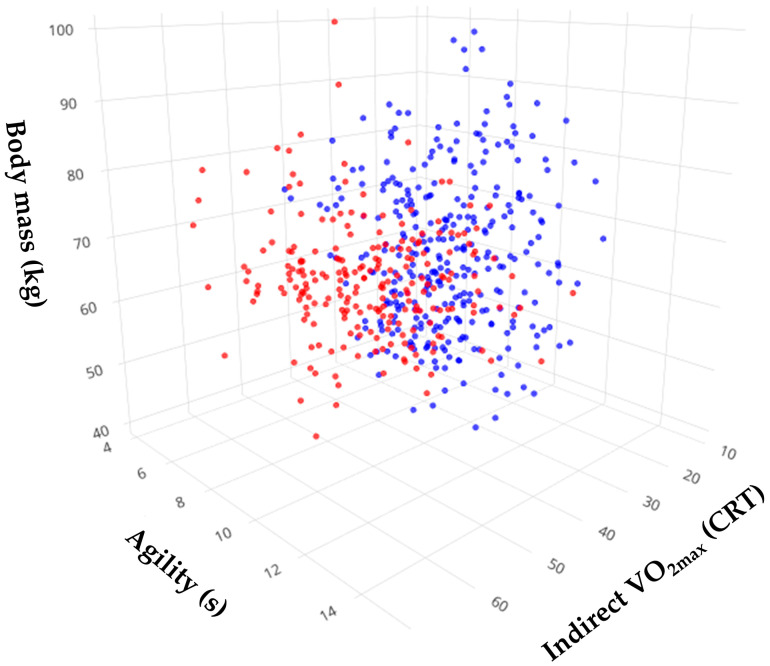
Multidimensional 3D plot of pre-defined variables. The arrangement of clusters within a 3D structure enables to identify of the significant variables that contribute the most to data variation by the natural generation of subspaces.

**Table 1 ijerph-20-00146-t001:** Characteristics of the participants.

Variable	Male (n = 445)	95% CI	Female (n = 97)	95% CI	*p*-Value	η^2^
Age	20.0 (2.2)	19.7–20.2	19.4 (2.2)	18.9–19.8	<0.001	0.013
BM (kg)	67.8 (9.8)	66.9–68.77	57.8 (8.6)	56.0–59.5	<0.001	0.144
Stature (cm)	171.7 (6.2)	171.2–172.3	159.2 (5.9)	158.0–160.4	<0.001	0.324
BMI (kg·m^−2^)	22.9 (2.7)	22.6–23.2	22.7 (2.9)	22.1–23.3	0.218	0.002

Data are expressed as mean (SD). BM: body mass; BMI: body mass index. The statistically significant differences at a level of 0.05 for the Mann–Whitney U test are shown. Effect size as eta-squared (η^2^).

**Table 2 ijerph-20-00146-t002:** Characteristics of the participants grouped by academic year and socioeconomic strata.

**Variable**	**Freshman**			**Sophomore**			**Junior**		**Senior**
	**(n = 279)**			**(n = 132)**			**(n = 59)**		**(n = 72)**
Age	18.8 (1.8) ^a,b,c^			20.5 (2.1) ^e^			20.2 (1.3) ^f^		22.3 (2.2)
Body mass (kg)	65.9 (10.4)			66.0 (10.2)			65.7 (9.8)		66.8 (10.4)
Stature (cm)	169.8 (8.1)			169.0 (7.0)			170.9 (8.7)		169.8 (7.2)
BMI (kg·m^−2^)	22.9 (2.7)			23.0 (3.0)			22.4 (2.2)		23.1 (2.7)
CRT Distance covered (km)	2.30 (0.4) ^c^			2.33 (0.4) ^e^			2.20 (0.4)		2.09 (0.4)
CRT VO_2max_	40.1 (9.5) ^c^			40.8 (10.5) ^e^			38.0 (10.1)		35.7 (10.0)
SRT-20M Final stage velocity (km·h^−1^)	11.8 (1.1) ^b^			12.0 (0.8) ^d^			12.8 (1.4) ^f^		11.6 (0.9)
SRT-20M VO_2max_	49.7 (6.5) ^b^			50.8 (5.0) ^d^			55.8 (8.4) ^f^		48.9 (5.7)
Horizontal jump (cm)	195.4 (30.1)			202.2 (34.6)			198.7 (21.7)		193.2 (32.3)
Sprint 30 m (s)	8.0 (2.0) ^a,c^			6.5 (2.2) ^d^			7.8 (0.7) ^f^		6.6 (2.0)
SRAT (s)	9.1 (2.1) ^a,b^			10.1 (1.4) ^e^			10.2 (0.7) ^f^		9.2 (1.3)
Sit and reach test (cm)	4.9 (8.3)			5.3 (9.5)			9.2 (8.1)		5.2 (8.6)
**Variable**	**SS1** **(n = 2)**		**SS2** **(n = 89)**		**SS3** **(n = 347)**		**SS4** **(n = 90)**	**SS5** **(n = 10)**	**SS6** **(n = 4)**
Age	18.5 (2.1)		19.2 (2.0) *		19.9 (2.2)		20.2 (2.3)	19.8 (1.3)	21.7 (2.9)
Body mass (kg)	69.1 (1.5)		65.0 (10.9)		65.8 (9.9)		68.0 (11.6)	65.9 (6.7)	64.4 (7.0)
Stature (cm)	173.5 (2.1)		169.2 (8.9)		169.2 (7.3)		171.0 (8.3)	166.7 (6.1)	169.5 (7.0)
BMI (kg·m^−2^)	22.9 (0.0)		22.6 (2.7)		22.9 (2.7)		23.1 (2.7)	23.8 (3.0)	22.4 (2.6)
CRT Distance covered (km)	2.67 (0.1)		2.32 (0.5)		2.25 (0.4)		2.27 (0.3)	2.41 (0.4)	2.39 (0.04)
CRT VO_2max_	48.5 (2.3)		40.6 (11.1)		39.0 (10.2)		39.5 (8.4)	42.6 (9.5)	42.1 (0.8)
SRT-20M Final stage velocity (km·h^−1^)	12.7 (1.0)		12.2 (1.3)		11.9 (1.0)		11.7 (1.0)	11.5 (0.8)	11.1 (0.4)
SRT-20M VO_2max_	55.2 (6.2)		52.0 (7.8)		50.5 (6.4)		49.5 (6.3)	47.8 (5.1)	45.7 (2.8)
Horizontal jump (cm)	229.5 (17.6)		195.9 (33.4)		196.7 (30.4)		197.9 (27.3)	208.8 (40.1)	203.2 (62.5)
Sprint 30 m (s)	5.3 (2.3)		7.6 (2.2)		7.5 (2.0)		7.3 (2.1)	7.2 (2.6)	7.6 (1.4)
SRAT (s)	10.5 (0.9)		9.5 (1.9)		9.4 (1.7)		9.4 (2.1)	10.1 (1.3)	10.3 (0.6)
Sit and reach test (cm)	4.5 (3.5)		5.6 (8.1)		5.4 (8.7)		5.3 (9.0)	9.1 (10.1)	14.7 (2.8)

Data are expressed as mean (SD). The social stratification in Colombia includes six strata, as follows: SS1 is lower-low, SS2 is low, SS3 is upper-low, SS4 is medium, SS5 is medium-high, and SS6 is high. Estimated VO_2max_ expressed in mL·kg^−1^·min^−1^. CRT: Cooper Run Test; SRAT: Shuttle Run Agility Test; SRT-20m: Shuttle Run Test; VO_2max_: Maximum oxygen uptake. *p*-value < 0.05 for the Kruskal–Wallis test (Bonferroni post hoc). ^a^ Difference between freshman and sophomore; ^b^ difference between freshman and junior; ^c^ difference between freshman and senior; ^d^ difference between sophomore and junior; ^e^ difference between sophomore and senior; ^f^ difference between junior and senior; * difference between SS2 and SS4.

**Table 3 ijerph-20-00146-t003:** Characteristics of the profiled phenotypes.

Variable	Cluster 1 (n = 318)	95% CI	Cluster 2 (n = 224)	95% CI	*p*-Value	η^2^
Age (years)	20.1 (2.5)	19.8–20.4	19.5 (1.6)	19.2–19.7	<0.001	0.008
Body mass (kg)	67.6 (11.4)	66.3–68.9	63.7 (8.0)	63.5–64.8	<0.001	0.031
Stature (cm)	169.1 (8.5)	168.1–170.0	170.1 (6.5)	169.2–171.0	0.559	0.000
BMI (kg∙m^−2^)	23.2 (2.9)	23.2–23.8	22.0 (2.2)	21.7–22.2	<0.001	0.075
CRT Distance covered (km)	2.10 (0.5)	2.06–2.15	2.50 (0.3)	2.45–2.55	<0.001	0.174
CRT VO_2max_	35.8 (9.3)	34.8–36.9	44.6 (8.7)	43.4–45.7	<0.001	0.174
SRT Final stage velocity (km·h^−1^)	11.3 (0.8)	11.2–11.4	12.8 (0.9)	12.6–12.9	<0.001	0.418
SRT VO_2max_	47.0 (5.0)	46.5–47.6	55.5 (5.4)	54.8–56.2	<0.001	0.418
Horizontal jump (cm)	194.6 (32.8)	191.0–198.2	200.7 (27.6)	197.1–204.3	<0.001	0.008
Sprint (s)	6.97 (2.1)	6.73–7.21	8.21 (1.8)	7.97–8.45	<0.001	0.085
SRAT (s)	10.1 (1.4)	9.97–10.29	8.65 (1.9)	8.39–8.91	<0.001	0.162
Sit and reach test (cm)	5.02 (8.8)	4.04–6.00	6.34 (8.4)	5.23–7.46	0.115	0.004

Data are expressed as mean (SD) with the corresponding 95% confidence interval (95% CI). CRT: Cooper Run Test; SRAT: Shuttle Run Agility Test; SRT-20M: Shuttle Run Test; VO_2max_: maximum oxygen uptake. The statistically significant differences at a level of 0.05 for the Mann–Whitney U test are shown. Effect size as eta-squared (η^2^).

## Data Availability

The data supporting the findings of this study are available from the corresponding author upon request.

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
