# Peer review of "Profiling Physical Fitness of Physical Education Majors Using Unsupervised Machine Learning"

_ijerph, 2022, doi:10.3390/ijerph20010146_

Round 1
Reviewer 1 Report
GENERAL
This manuscript aimed to profile the physical fitness of physical education majors using unsupervised machine learning and to identify differences according to sex, academic experience and socioeconomic strata.
In general, this is an interesting and well-conducted study that may be useful to complement the available literature in this research line. However, there are some aspect to attend. Some of the most important aspects concern to methodological information that make this study non-replicable. At this moment, some parts of this section need to be improved by adding detailed information and references and/or data that support these procedures. Also, discussion may be improved by incorporating interpretation of results and practical applications. Finally, fluency and clarity of the text should be checked, as well as some points related to data in results and tables.
These aforementioned aspects are detailed by sections:
Abstract
Some abbreviations without explaining should be checked (i.e. F, M, BMI…).
It would be appropriate to incorporate some statistical data to abstract.
Introduction
Text has several run-on sentences that make it difficult to follow. Please, check it:
- lines 49-53
- lines 54-57
- lines 58-61
- lines 70-73
Methods
Maybe, it would be useful to have a supplementary file with the STROBE checklist.
Recommendations given to participants should be supported by previous similar studies.
How was the recruitment process? Was a private institution randomly chosen?
Familiarization process is supported by any previous study?
The time of data collection was between 07:00 to 10:00 and 132 18:00 to 21:00, according to the time when participants were available. However, this could have affected results. How was it controlled?
Again, references to support validity and reliability of data collection procedures and tools should be included for all the test.
Sprint speed: what tool was used to register this data? Authors state “the time data obtained during the execution was recorded”, but… how? what procedures? validity?
It is needed that authors increase information about how each test was carried out (mainly sprint speed, agility and flexibility). Now, these test are not detailed enough to be replicated.
Sample size is not justified. Please, detail the sample size calculation.
How many studies replied to the announcement? Maybe, it would be useful a flow diagram of the participants.
Statistical analysis is recommended that is distributed in an only paragraph.
Results
As I aforementioned, flow diagram with reasons is highly recommended.
Why 71 participants did not complete all the tests?
I think authors are categorizing according to sex (female and male; sex is biological attributes in humans and animals that is most often associated with physical and physiological features of an individual). So, this should be modified in the entire manuscript (instead of women and men, which is gender; Gender refers to as the socially constructed role, behaviour, expression or identity of an individual (ie, girls, boys, women, men, gender diverse).
Tables (1, 2 and 3) should be understandable by itself. Now, there are several abbreviations without explaining that should be attended. Please, review it.
Authors should include some statistical data referred to effect sizes, and also to complete Table 2 or text with 95%CI, p value, effect sizes…
Discussion
Interpretation of results should be improved in order to clarify the importance of your results in contrast with the available literature.
Interpretation: this part is a run-on sentence very difficult to understand. Please, re-arrange.
Conclusion is too large and confusing (i.e. lines 137-138 belong to discussion body). Please, shorten this section and increase discussion to interpret data and results. Also, it would be needed to separate and improve practical application derived from your study.
Author Response
Dear reviewer,
Please find attached the point-by-point answers to your review report.
Sincerely,
The authors

Reviewer 2 Report
Profiling Physical Fitness of Physical Education Majors using Unsupervised Machine Learning
The study aimed to identify whether physical fitness of physical education majors may affected by their year of study, gender and socioeconomic strata. The study is well written, the methods are well described and the results are solid. The discussion however, needs improvements since many significant results from the study are not presented or discussed. I comment the Authors for the large number of participants. The manuscript is in a very good state, thus I have only minor comments.
1. Abstract: Abstract is well written and provides a good overview of the study.
Line 28: Change to 445Males, 97Females and adage, body mass and body height.
2. Introduction: Introduction presents nicely the research question and the problem of the study. Has a good flow and relative references. No comments here.
3. Methods: I have some questions and suggestions for methods:
General: How many trials were allowed for all performance tests? What value was used for the statistics, the best performance or the mean performance? What was the time rest between trials? Were there any special instructions in field measurements? What measuring instruments Authors used for performance measurements?
Paragraph Settings: What was the exact order of measurements? How many times the participants visited the laboratory and field?
Participants: I suggest providing a reference here to Table 1 for age and anthropometric characteristics of the participants.
Line 124: How the socioeconomic stratum was measured?
Data sources/measurement: So if I get this right, all subjects participated into two testing sessions’ one familiarization and one either in the morning or in the afternoon? Normally, during noon hours performance is more elevated compared to morning hours. Maybe a reference here will help Authors for this experimental decision. In addition, was only one familiarization session enough for participants?
Lower-Limb Strength/Power: Why the horizontal jump is a strength measurement? Horizontal jump is a lower body power test (doi: 10.1519/JSC.0b013e3181739838, DOI:10.7752/jpes.2019.s2064).
Sprint Speed and Agility: How the sprint test time and agility time were measured?
Study size: Is the number 613 a result from power analysis?
Statistical analysis: Why non-parametric statistics were used?
4. Results: Results are clearly presented, tables and figures are solid. Just a small notice for table 2: Authors should clarify how the levels of socioeconomic strata were created.
5. Discussion: Discussion is too small. Authors have done a great scientific work with many results and findings. There is plenty room for presenting and discussing the results.
Limitations: Is there a record for athletic status of the participants? This might be a limitation as well.
Well done!
Author Response

(The authors gave the same response as above.)

Round 2
Reviewer 1 Report
This is a revised paper about physical fitness profiles of physical education majors using unsupervised machine learning and to identify differences according to sex, academic experience and socioeconomic strata.
I congratulate the authors, they have made efforts to include a lot of revisions based on the reviewers' comments. These changes have greatly improved the quality of the report. I feel that the manuscript provides good evidence for the use of these findings in exercise recommendations of low physical performance and overweight majors.
Reviewer 2 Report
no comments